# TDP2 drives immune evasion and metastatic progression in prostate cancer

Huan Cao[1☯*], Anyin Pan[2☯], Yuetao Chen[2], Fei Zheng[2]

1 Haining People's Hospital, Haining, Zhejiang, P.R. China, 2 Sanatorium Zone 3, Hangzhou Special Service Sanatorium Center, Hangzhou, Zhejiang, P.R. China

☯ These authors contributed equally to this work.
* happygrass001@sohu.com

## Abstract

Immune evasion and epithelial-mesenchymal transition (EMT) are critical mechanisms driving tumor progression and therapy resistance in prostate cancer. In this study, we explored the role of TDP2 in modulating the tumor microenvironment (TME) through single-cell RNA sequencing and pathway enrichment analysis. Our results revealed that epithelial cells with high TDP2 expression extensively interact with myeloid cells, macrophages, and fibroblasts, thereby shaping immune responses and facilitating tumor progression. Specifically, TDP2 overexpression suppressed M1 macrophage polarization and dendritic cell (DC) maturation, leading to reduced CD8 + T cell activation and enhanced immune evasion. Additionally, TDP2-high expression was associated with enriched signaling pathways involved in EMT, including COLLAGEN, GALECTIN, MIDKINE (MK), and ONCOSTATIN M (OSM), which promoted tumor cell migration, invasion, and immune evasion. Survival analyses further demonstrated that high TDP2 expression correlated with poor clinical outcomes in prostate cancer patients. Overall, our findings identify TDP2 as a key regulator within the TME and suggest its potential utility as both a prognostic biomarker and therapeutic target in prostate cancer.

## Introduction

Prostate cancer (PRAD) is one of the most common malignancies in men worldwide, representing a major contributor to global cancer morbidity and mortality [1]. Despite advancements in therapeutic strategies, including surgery, radiation, and androgen deprivation therapy, many patients eventually develop resistance to treatment, leading to disease progression and metastatic spread [2]. Immunotherapy has emerged as a promising approach for cancer treatment; however, its efficacy in PRAD remains limited, primarily due to the highly immunosuppressive tumor microenvironment (TME) and the lack of well-characterized immune targets [3]. Thus, identifying novel immune-related targets is critical to address these limitations and improving therapeutic outcomes for PRAD patients.

**Data availability statement:** The data that support the findings of this study are available from the Gene Expression Omnibus (GEO) database (accession ID: GSE181294; https://www.ncbi.nlm.nih.gov/geo/query/acc.cgi?acc=GSE181294) and The Cancer Genome Atlas (TCGA) database (project: TCGA-PRAD; https://portal.gdc.cancer.gov/). The data supporting the findings of this study are also available upon reasonable request from the institutional contact at Haining People's Hospital (email: info@hnph.com). Data access will be provided in accordance with institutional and ethical guidelines.

**Funding:** The author(s) received no specific funding for this work.

**Competing interests:** The authors have declared that no competing interests exist.

The tumor immune microenvironment plays a pivotal role in prostate cancer progression and therapy resistance. It is composed of diverse immune cell populations, including tumor-associated macrophages (TAMs), T cells, and myeloid-derived suppressor cells (MDSCs), which dynamically interact with cancer cells to either facilitate or inhibit tumor growth [4]. Recent advances in single-cell RNA sequencing (scRNA-seq) have elucidated unprecedented insights into the cellular heterogeneity and molecular dynamics of the TME, enabling the identification of key regulators of tumor-immune interactions [5]. Utilizing these technologies, researchers can uncover novel therapeutic targets capable of reshaping the immunosuppressive landscape of PRAD and enhancing anti-tumor immunity.

TDP2 (Tyrosyl-DNA phosphodiesterase 2), a DNA repair enzyme, has recently attracted increasing attention for its potential role in cancer biology. Initially identified for its ability to repair DNA damage induced by topoisomerase inhibitors, TDP2 is known to maintain genomic stability and modulate cellular stress responses [6]. Through integrated analysis of gene expression data from The Cancer Genome Atlas (TCGA) and scRNA-seq datasets, TDP2 has been identified as a potential prognostic factor in PRAD. Emerging evidence further suggests that TDP2 may regulate immune-related signaling pathways, such as the p38 MAPK pathway, which is critical for immune cell activation and cytokine production [7]. Notably, tumor cells with elevated TDP2 expression exhibit increased interactions with components of the TME, potentially suppressing anti-tumor immunity and facilitating tumor progression. These observations underscore the need to investigate the role of TDP2 in immune modulation and its potential as a therapeutic target in PRAD.

In this study, we aimed to characterize the functional significance of TDP2 in prostate cancer, with a focus on its impact on the immune microenvironment. By integrating bulk RNA sequencing data from TCGA and scRNA-seq analysis, we investigated the association between TDP2 expression and immune-related pathways, as well as its role in mediating tumor-immune interactions. Our findings highlight TDP2 as a crucial regulator of immune suppression in PRAD and provide a rationale for targeting TDP2 to enhance anti-tumor immunity and improve patient outcomes.

## Materials and methods

### The collection and analysis of data

We collected gene expression data from a total of 492 prostate cancer (PRAD) patients and 52 normal prostate tissues, and their corresponding clinical information from the TCGA database (https://portal.gdc.cancer.gov/). Additionally, single-cell RNA sequencing (scRNA-seq) data (accession ID: GSE181294) were obtained from the GEO database (https://www.ncbi.nlm.nih.gov/geo/), including samples from 18 prostate tumors, 14 adjacent normal tissues, and 5 healthy prostate tissues. Cells containing fewer than 600 unique molecular identifiers (UMIs) were excluded as low-quality. To remove doublets, Scrublet was applied, and cells with doublet scores exceeding 0.4 were filtered out.

Single-cell data were further processed using Seurat v4. Principal component analysis (PCA) was conducted on individual cell samples, identifying the top 20 principal components (PCs). These components were then chosen for further study. The UMAP algorithm was employed to comprehensively analyze dimensionality reduction on the initial 20 PCs.

### Identification of differentially expressed genes (DEGs)

Differential expression analysis comparing tumor and normal prostate tissues was performed using the "limma" R package [8]. Differentially expressed genes (DEGs) were identified with criteria of adj. $P < 0.05$ and $|Log2FC| > 1$. The results were visualized using volcano plots and the overlap between DEGs was displayed via Venn diagrams.

### Screening key biomarkers

We focused on epitheial cells 2, distinguishing tumor, adjacent, and healthy tissue samples. Through differential gene analysis, we identified PRAD cancer cell-related genes with the standard: adj. $P < 0.05$ and $|Log2FC| > 1$. Next, we used the TCGA-PRAD data to perform differential analysis between cancer and adjacent tissues, and screened out differentially expressed genes using adj. $P < 0.05$ and $|Log2FC| > 1$ as thresholds. We intersected the differentially expressed genes with the same trend, and these genes were considered to be important genes associated with the occurrence of PRAD. Subsequently, LASSO regression ("glmnet" package) and multivariate Cox regression ("survival" package) were employed to identify prognostic biomarkers.

### Survival analysis

Survival analysis was performed using the TCGA gene expression and clinical data on the GEPIA2 [9] web server (http://gepia2.cancer-pku.cn/).

### Functional enrichment analyses

Gene Ontology (GO), Kyoto Encyclopedia of Genes and Genomes (KEGG), and Hallmark pathway enrichment analyses were conducted using the R package clusterProfiler. Pathway enrichment results with $p < 0.05$ were considered significant, and Gene Set Enrichment Analysis (GSEA) was used to evaluate specific gene sets under different conditions.

### Cell–cell communication analysis

Cellular communication was analyzed using the CellChat R toolkit [10], where ligand-receptor interactions were inferred from gene expression data to assess cell-cell communication.

### Cell lines

293T (HEK 293T), 22RV1, RM-1, PC3 and C4-2 were obtained from ATCC. 293T and RM-1 were cultured in DMEM/high glucose medium (C11995500BT, Thermo Fisher Scientific) containing 10% FBS (A5669701, Thermo Fisher Scientific) plus 1% penicillin/streptomycin (E607011, Sangon Biotech (Shanghai) Co. Ltd.). 22RV1, PC3 and C4-2 were cultured in RPMI-1640 medium (C11875500BT, Thermo Fisher Scientific) containing 10% FBS plus 1% penicillin/streptomycin. All cells were maintained at 37°C incubator with 5% $CO_2$. Cells were used for experiments within 10–20 passages from thawing. All cells were authenticated via short tandem repeat testing. All cells were routinely tested for ensuring mycoplasma free.

### Antibodies and reagents

Mouse anti TDP2 (sc-515179), mouse anti ERK 1/2 (sc-514302), mouse anti JNK (sc-7345), mouse anti JNK Phospho-Thr18/Tyr185 (sc-6254) were from Santa Cruz. Rabbit anti Erk1/2 Phospho-Thr202/Tyr204 (4370), rabbit anti

p38 (8690), rabbit anti p38 Phospho-Thr180/Tyr182 (4511), rabbit anti CD8α (98941),rabbit anti CD3ε (78588), rabbit anti Granzyme B (44153) were from Cell Signaling Technology. Rabbit anti GAPDH (81640–5-RR) was from Proteintech. PE-CF594 rat anti-CD11b (562399), PE rat anti-CD86 (561963), Alexa Fluor 647 rat anti-CD206 (565250), BV421 rat anti- I-A/I-E (562564), APC-R700 rat anti-CD103 (565529), PE-Cy7 rat anti-CD8a (552877), BV786 hamster rat anti-CD69 (564683) and Purified rat anti-mouse CD16/CD32 (553141) were from BD Pharmingen. CD8+ T cell isolation kit (130-096-495) was from Miltenyibiotec. Red blood cell lysis buffer (abs9101) and LPS (abs42020800) was from ABsin. Mouse M-CSF Recombinant Protein (KPRT0174), Mouse GM-CSF Recombinant Protein (KPRT0153), Mouse IL-4 Recombinant Protein (KPRT0162) were from Genie.

## Overexpression and depletion of genes in cell lines

Stable overexpression of TDP2 was accomplished in 22RV1, RM-1, PC3 and C4-2 cells Full-length expression cDNA of TDP2 (human and mouse) plasmids were purchased from Sino Biological Inc. TDP2(h) shRNA plasmid and TDP2(m) shRNA plasmid were purchased from Santa Cruz Biotechnology, Inc. The plasmids were co-transfected with psPAX2 and pMD2.G into 293T cell using Lipofectamine 2000. The supernatant containing packaged lentiviral particles was collected, centrifuged, mixed with polybrene, and added into 22RV1, RM-1, PC3 or C4-2 cells respectively. After 48 h of infection, the cells were selected in a medium containing 1 µg/ml puromycin for 14 days to generate the stable cell lines.

## Westernblot analysis

Cells were lysed in RIPA buffer with protease inhibitors (Beyotime Biotechnology) containing proteinase inhibitor cocktail (Beyotime Biotechnology). as described before [11]. The protein concentration in the supernatant was determined using Bio-Rad protein assay reagent. Protein samples of cell lysates were separated by 10–12% SDS–PAGE, and then trans-ferred onto nitrocellulose membrane (Millipore) following standard procedures. Membranes were blocked with 5% BSA in TBST for 1 hour and subsequently incubated with indicated primary antibodies overnight at 4°C. After three washes with TBST, membranes were incubated with HRP-conjugated secondary antibodies for 1 hour at room temperature. The mem-branes were visualized using a ChemiScopeTouch imaging system. The intensity of proteins expression was quantified using Image J.

## Cell proliferation assay

Cells from different experimental groups were plated in fresh 6-well plates at a density of $2 \times 10^5$ cells per well. The cell numbers were subsequently quantified on days 2, 4, and 6 post-seeding to compare growth rates.

## Colony formation assay

Approximately $1 \times 10^3$ cells per group were seeded on 12-well plates in triplicate and incubated for 10–14 days with regular medium changes every three days to support colony growth. After the incubation period, colonies were fixed in absolute methanol for 30 min, and stained with 0.1% crystal violet for another 15 min. Subsequently, the plates were washed and examined for colony visualization.

## Transwell migration assay

Migration capacity of prostate cancer (PC) cells was evaluated via the transwell assays with transwell compartments. Following transfection, $3 \times 10^4$ 22RV1 cells or RM-1 cells were seeded in the upper chamber of transwell units (NEST) with 8 µm pore size polycarbonate filters under serum-free conditions. And the lower chamber was filled with 600 µL RPMI-1640 or DMEM/high glucose medium containing 10% FBS. After incubation for 24 hours under 37°C, the migrated cells were fixed with 4% paraformaldehyde and stained with 0.1% crystal violet solution. Imaging was done with an inverted

microscope (Olympus) with a 20×objective. Images of five fields per insert were captured. The migrated cells stained with crystal violet were quantified and compared.

## Wound-healing assay

After transfection, the cells were cultured in complete medium in 6-well plates until reaching approximately 100% confluence. Subsequently, A vertical scratch was created using a 1000 μL pipette tip, followed by washing the wells with phosphate-buffered saline and replacing the medium with new serum-free medium. Images were captured at 0 h and 24 h post-scratching with an inverted microscope (Olympus) with a 5×objective. Wound closure percentage was calculated using the formula: % wound closure = ((0 h wound area – 24 h wound area) × 100)/0 h wound area.

## Transplant tumor formation assay

C57BL/6J mice (Male 4 weeks old) and Nude mice (Male 4 weeks old) were obtained from Hangzhou Medical College. RM-1 or RM-1 TDP2 KD cells ($5\times10^5$ cells suspended in 50 μl PBS) were injected into the right flank of 6-week-old male C57BL/6J mice or Nude mice to establish the subcutaneous implantation model. Tumor size was monitored by a caliper, and tumor volume was calculated using the formula: $length \times width^2 \times 0.5$. The ethical guidelines limited the tumor size to 2000 mm$^3$ per mouse, and this threshold was strictly adhered to throughout the study. Cell implantation was performed under isoflurane anesthesia to minimize discomfort.

At the end of the experiment, mice were euthanized by gradual-fill $CO_2$ asphyxiation in accordance with institutional animal welfare protocols. Mice were placed in the euthanasia chamber before $CO_2$ administration. $CO_2$ was then introduced at a rate of approximately 30% of the chamber volume per minute. After confirming the cessation of respiration and heartbeat, $CO_2$ was turned off, and mice were observed for an additional two minutes to ensure death. Carcasses were subsequently removed for downstream experimental procedures. Upon sacrifice, the tumors were excised, weighed, and preserved in 10% formaldehyde for further analysis.

## Animal welfare and humane endpoints

All animal experiments were approved by Animal Experimental Ethical Inspection of Hangzhou Special Service Sanatorium Center, ID of the approval: 20240121003. Animals were housed under specific pathogen-free (SPF) conditions with a 12 h light/dark cycle and ad libitum access to food and water.

Humane endpoints were defined and monitored throughout the experiment. Mice were observed at least twice daily for general health and tumor progression. Criteria for euthanasia included: (1) tumor volume exceeding 2000 mm³, (2) weight loss greater than 20% of baseline, (3) impaired mobility or inability to access food or water, (4) ulcerated or necrotic tumors, or (5) signs of severe distress such as hunched posture, labored breathing, or persistent lethargy. Once mice met endpoint criteria, euthanasia was performed within 6 hours using gradual-fill $CO_2$ asphyxiation followed by cervical dislocation, in accordance with AVMA guidelines.

In total, 8 mice were used in the transplantation experiments. The experiment lasted 17 days post-implantation. All animals were euthanized when tumors reached the size limit or when humane endpoint criteria were met. No animals were found dead before reaching the criteria. All procedures, including tumor implantation, monitoring, and euthanasia, were carried out by staff trained in animal handling and welfare to ensure the highest standards of care.

## Immunohistochemistry (IHC)

Formalin-fixed paraffin-embedded mouse tumor tissue sections were analyzed by IHC for CD3,CD8 and Granzyme B.Following the manufacturer's instructions, the slides with 4 μm-thick tissues,were baked for 1 hour at 68 °C, then deparaffinized. Antigen retrieval was performed using sodium citrate buffer (Solarbio). Samples were blocked in 3% BSA for 30

min at room temperature, incubated overnight with a primary antibody at 4°C. This was followed by incubate to a HRP-conjugated secondary antibody at room temperature for 40 min. Target proteins were visualized using a diaminobenzidine (DAB) Substrate Kit (Solarbio). Slides were counterstained with diluted hematoxylin for 3 min. Representative images of each tumor were captured using upright microscope (Olympus) with a 20 × objective. The immunohistochemical results were further quantified by Image J software.

### In vitro co-culture assay

In brief, the co-culture system consists of RM-1(WT or TDP2 OE), mice BMDMs, mice BMDCs, or mice spleen-derived T lymphocytes. RM-1 cells from different experimental groups were seeded into 6-well plates at a density of $1 \times 10^5$ per well and incubated 12 hours. To analyze the macrophage polarization, the isolated BMDMs were co-cultured with RM-1 WT cells or RM-1 TDP2 OE cells at the ratio of 1:1. After 48 hours incubation, the cells in the plate were collected and analyzed by flow cytometry. The BMDMs treated with LPS (0.5 μg/mL) for 48 hours were set as a positive control. The cells were incubated with CD11b, CD86, MHC II, and CD206 antibodies and the CD11b antibody was used to gate the BMDM group. To analyze the DC maturation, the isolated BMDCs were co-cultured with RM-1 WT cells or RM-1 TDP2 OE cells at the ratio of 1:1 for 48 hours. Then, the cells in the plate were analyzed by flow cytometry. The cells were incubated with CD11b and CD86 antibodies and the CD11b antibody was used to gate the BMDC group. The BMDCs treated with LPS (0.5 μg/mL) for 48 hours were set as positive reference. To analyze the CD8+ T cell maturation, a total of $2 \times 10^5$ T lymphocytes were incubated with cancer cells for 48 hours with the existence of BMDMs and BMDCs. Then the cells were collected and incubated with CD8, CD103 and CD69 antibody to analyze the maturation.

## Results

### The prostate TME characterized by single-cell transcriptomic analysis

Single-cell RNA-seq data from 18 human prostate tumors and 14 matched adjacent normal tissues, alongside 5 healthy prostate tissues, were processed. After quality control, 156,261 cells were retained for analysis, with an average of 4,223 cells per sample. Using PCA and UMAP, 16 distinct cell clusters were identified, with epithelial cells showing diverse expression patterns (Fig 1A and 1E). According to the cell source, we show cells from healthy samples, tumor samples, and adjacent samples (Fig 1A). Interestingly, epithelial cells were divided into two groups with different expressions. PRAD is a tumor type of epithelial carcinoma, so tumor cells exist in these two groups of epithelial cells. In addition, it is obvious that epithelial cells2 are significantly less in healthy samples. Fig 1B stacked graph more intuitively shows the proportion of different cell types in the three groups. We found that the proportion of macrophages was significantly different in tumor tissue and adjacent tissue (Fig 1C), but there was no significant difference between the healthy sample group and adjacent group. At the same time, the proportion of epithelial cells2 in healthy samples was significantly less than that in adjacent samples and tumor samples. We speculate that this may be a group of epithelial cells that are most directly affected by tumorigenesis, so it is also one of our subsequent focuses.

### Key biomarker screening

To identify differentially expressed genes (DEGs) in PRAD samples relative to normal prostate tissues, we used the "limma" R package to identify genes with significant differences in expression with the criteria of adj. $P < 0.05$ and |Log2FC| > 1. We then used a volcano plot to visually display the DEGs (Fig 1F). To find the most critical genes in the development of the disease, we combined single-cell data for analysis. We focused on epithelial cells 2 and distinguished cells from tumor samples and cells from adjacent or healthy samples by source. Through differential gene analysis, we identified PRAD cancer cell-related genes with the criteria of adj. $P < 0.05$ and |Log2FC| > 1. We overlapped these DEGs with those with the same trend obtained from TCGA-PRAD, and these genes were considered to be important genes associated with the development of PRAD.

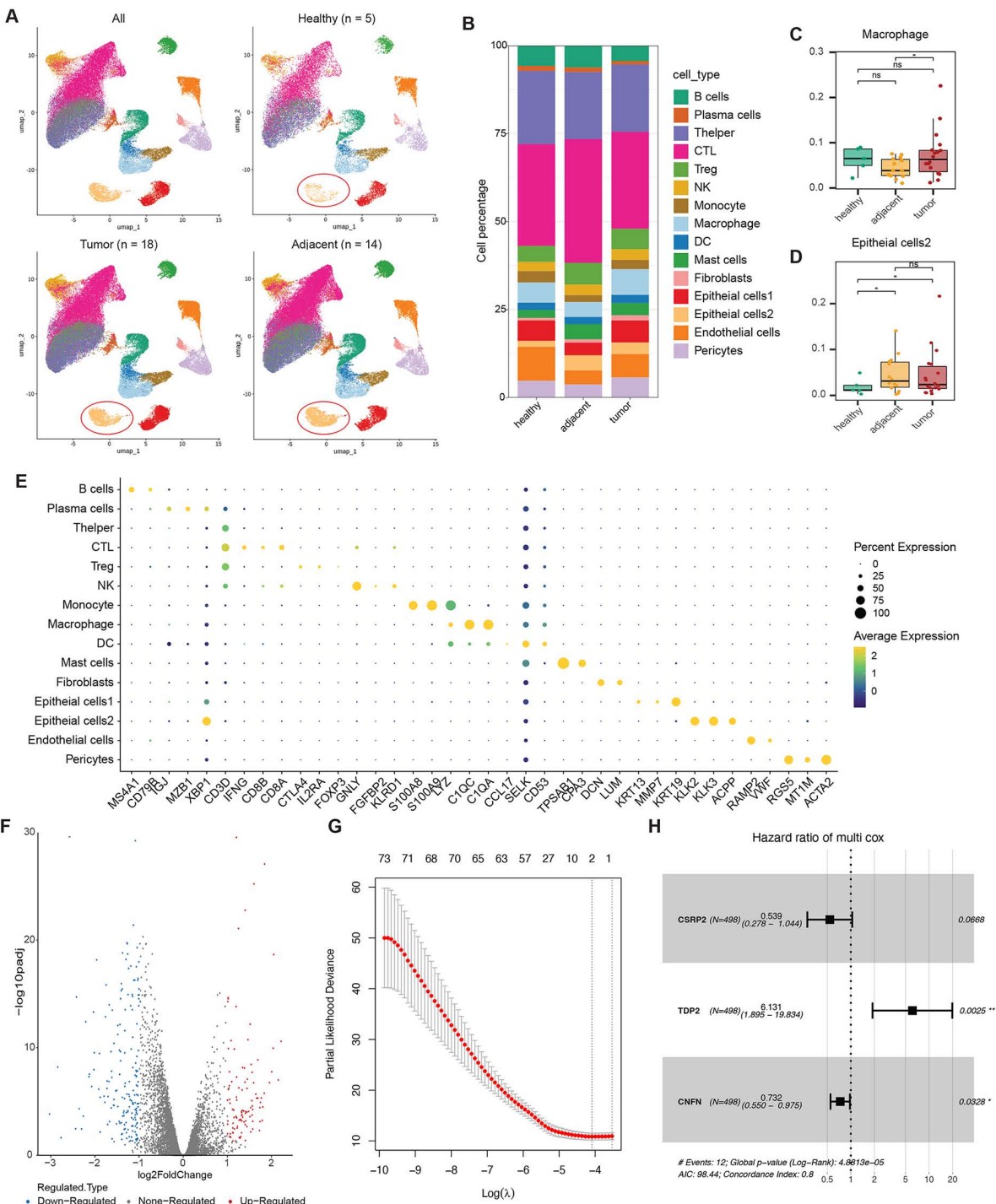

**Fig 1. Identification of key prognostic biomarkers in prostate cancer through scRNA-seq (GSE181294) and TCGA-PRAD Analysis. A** The cell type annotation of 156,261 cells using uniform manifold approximation and projection (UMAP) plots and split into individuals with 3 tissue origins. Colors indicate different cell types. **B** A stacked plot presenting the proportion of each cell type in the 4 tissue origins. Colors represent different cell types. Box plots showing the cell ratio of macrophages **(C)** and epithelial cells 2 **(D)** in the three groups. P values were determined by the student's t-test, *P < 0.05, ns Non-significant. **E** Bubble diagram of marker genes in each cell type. **F** The volcano plots of TCGA-PRAD DEGs. **G** Results of LASSO regression analysis. **H** Results of multivariate Cox regression analysis.

Combined with the prognostic data of TCGA-PRAD, we used Lasso regression (Fig 1G) and multivariate Cox regression to establish a prognostic prediction model (Fig 1G) to screen key biomarkers for prognosis. The importance of CNFN and TDP2 can be noted in both models. When the effects of these two genes on prognosis were examined separately, the Kaplan-Meier survival curve suggested that TDP2 had a stronger prognostic function (Fig 2A and 2B). Surprisingly, the feature plot in Fig 2C shows that TDP2 is specifically expressed in the epithelial cells 2 cluster, and not all epithelial cells 2 express TDP2. Therefore, the biological function of epithelial cells 2 with high expression of TDP2 deserves further exploration.

## Functional enrichment analysis of epithelial cells 2 with high expression of TDP2

We divided these cells into high TDP2 expression group and low TDP2 expression group based on the median TDP2 expression in epithelial cells 2. GO pathway enrichment analysis (Fig 2D) showed that the upregulated pathways associated with TDP2 high expression cells mainly involved immune response, cell proliferation and cell differentiation, which are closely related to the growth and metastasis of prostate cancer. KEGG pathway enrichment analysis (Fig 2E) further confirmed that TDP2 high expression cells were enriched in PPAR signaling pathway and fatty acid metabolism, suggesting that lipid metabolism plays an important role in the energy metabolism and survival of prostate cancer cells.

Hallmark pathway enrichment analysis (Fig 2F) revealed that classical signaling pathways associated with TDP2 high expression cells, such as TNFα signaling, KRAS signaling and hypoxia response, all play a driving role in the malignant transformation of the tumor microenvironment, which may promote tumor invasiveness and drug resistance.

The ssGSEA analysis (Fig 2G–2I) showed that TDP2-high-expressing cells were enriched in important biological processes such as negative regulation of inflammatory response (Fig 2G), extracellular matrix remodeling (Fig 2H), and macrophage differentiation (Fig 2I), emphasizing the close relationship between TDP2-high expression and immunosuppression, myeloid immune response, and mesenchymalization of the tumor microenvironment.

Overall, this part of the results reveals the different roles of prostate cancer epithelial cells with high and low TDP2 expression in tumor occurrence and development, especially their differences in immunosuppression, epithelial-mesenchymal transition (EMT), and myeloid immune response.

## TDP2 promote tumor progression by affecting the phenotype of PC

To reveal the important effect of TDP2 in PC cell lines, we established cell lines with either overexpression (OE) or knockdown (KD) of TDP2 in human prostate cancer cell lines (22RV1, PC3, C4-2) and a mouse prostate cancer cell line (RM-1). After transfection, the TDP2 expression levels of the cells were measured by Western blot analysis (Figs 4A, 4C, S1A, S1B and S9). Thus, the results indicated that the TDP2 overexpression (OE) or knockdown (KD) prostate cancer cell line had been successfully established.

We further investigated the impact of TDP2 on the growth of prostate cancer through cell growth experiments. We observed that transfection with shRNA-TDP2 significantly suppressed the proliferation of the prostate cancer cell lines compared to that of the cells in the blank control group, and transfection with pLV3-TDP2 plasmids inconspicuously promoted the proliferation of the prostate cancer cell lines (Figs 3A, 3B, S1C and S1D). Consistently,the colony formation assay showed that knockdown or overexpression of TDP2 affect the total number of colonies generated of prostate cancer cells(Figs 3C, 3D and S1E–S1J).

## TDP2 affect the adhesion ability of prostate cancer cells

To further investigate the impact of TDP2 on the biological functions of prostate cancer. We hypothesized that it might affect the adhesion ability of cells, thereby influencing the abilities of prostate cancer cells to migrate. Wound-healing and Transwell tests were used to assess the effects of TDP2 on prostate cancer cell migration. We observed that knocking down TDP2 significantly inhibited the migration ability of prostate cancer cells, while overexpression of TDP2 significantly enhanced the migration ability (Figs 3E, 3F, S2A and S2B).

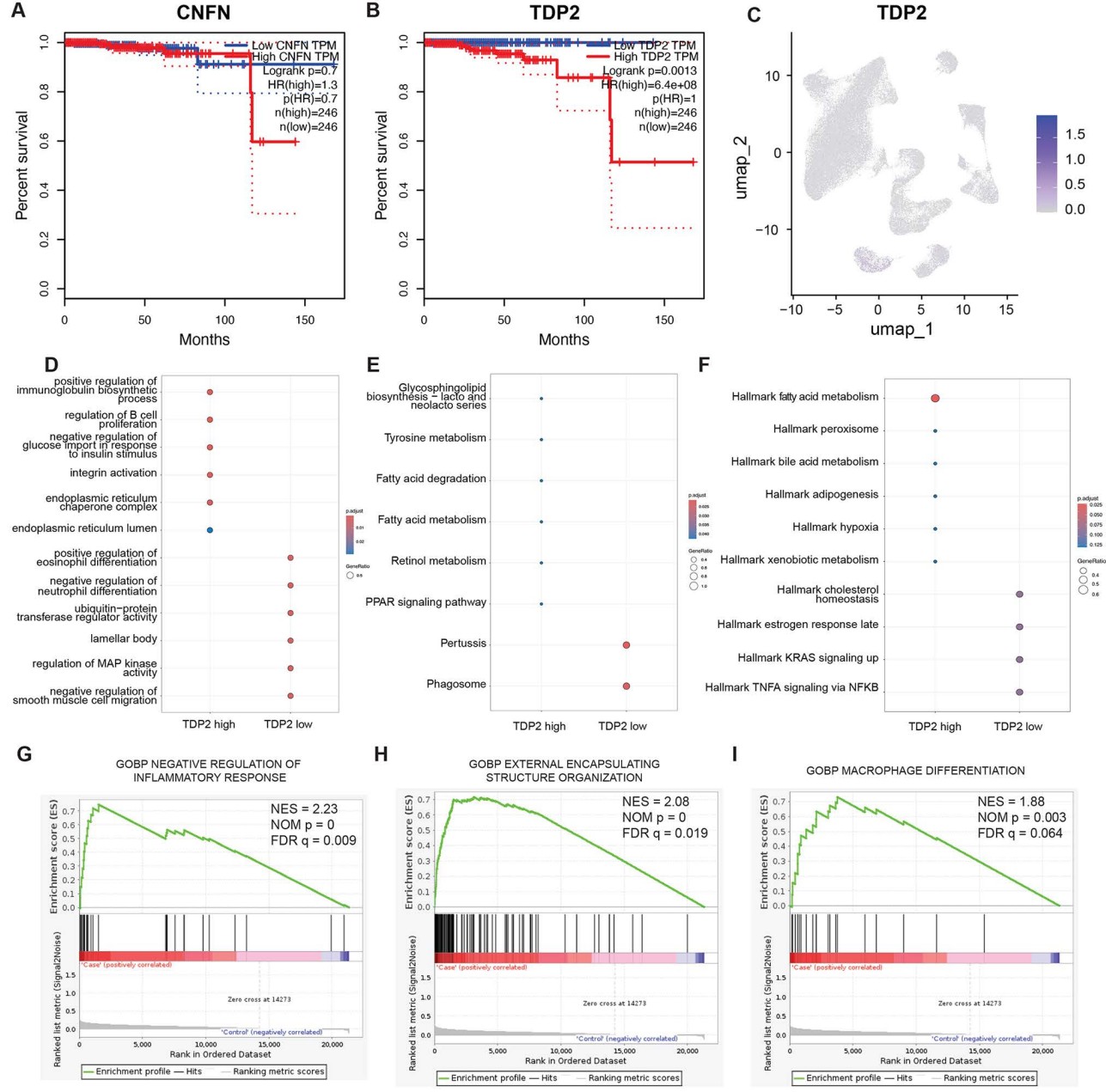

**Fig 2. Study on the mechanism of poor prognosis of tumor cells with high expression of TDP2.** The Kaplan-Meier survival curve of the CNFC **(A)** and TDP2 **(B)** in TCGA-PRAD. **C** UMAP visualization showing the expression of TDP2. **D** GO pathway enrichment analysis. **E** KEGG pathway enrichment analysis. **F** Hallmark pathway enrichment analysis. **G**, **H**, **I** Pathways enriched in TDP2 high cells in ssgsea analysis.

## TDP2 exerts biological functions through the MAPK pathway

Further explore the regulatory mechanisms of TDP2. Mitogen activating protein kinases (MAPKs) families, including c-Jun N terminal kinases (JNK), extracellular signal-regulated kinase (ERK), and MAPK p38 highly expressed in prostate cancer resulting in proliferation and invasion [12]. In the current study, JNK, p38, and ERK1/2 protein phosphorylation

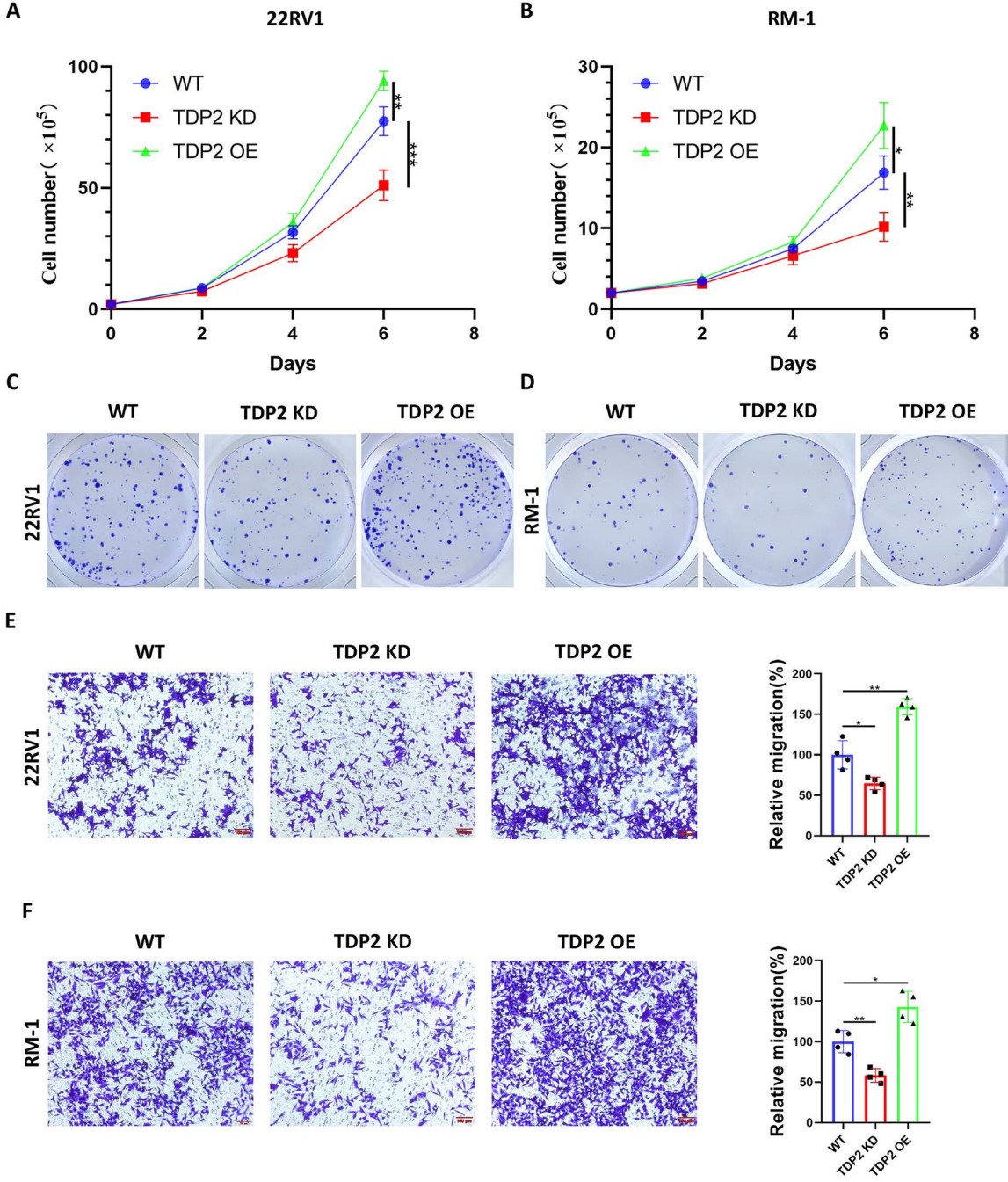

**Fig 3. TDP2 regulates proliferation and migration of prostate cancer cells.** Biological functions of TDP2 expression in 22RV1 and RM-1 cells. Cell growth curves **(A, B)** and colony formation assays **(C, D)** were performed to assess the proliferation of TDP2 KD or TDP2 OE transduced 22RV1 and RM-1 cells. Transwell assays in 22RV1 **(E)** and RM-1 **(F)** cells with or without TDP2 KD or OE was performed to assess the migration capacity. WT, wild type; TDP2 KD, TDP2 knockdown; TDP2 OE, TDP2 overexpression. Data were presented as means ± standard deviations. Error bars designate SD acquired from four independent experiments. P values were determined by the one-way ANOVA, *P < 0.05, ns Non-significant.

in prostate cancer cells 22RV1 and RM-1 were examined by western blotting. We observed that knockdown of TDP2 in 22RV1 and RM-1 significantly inhibited the phosphorylation level of p38, while overexpression of TDP2 sensibly promoted the overexpression of phosphorylated p38(Fig 4A–4D). But knockdown or overexpression of TDP2 has no effect on the

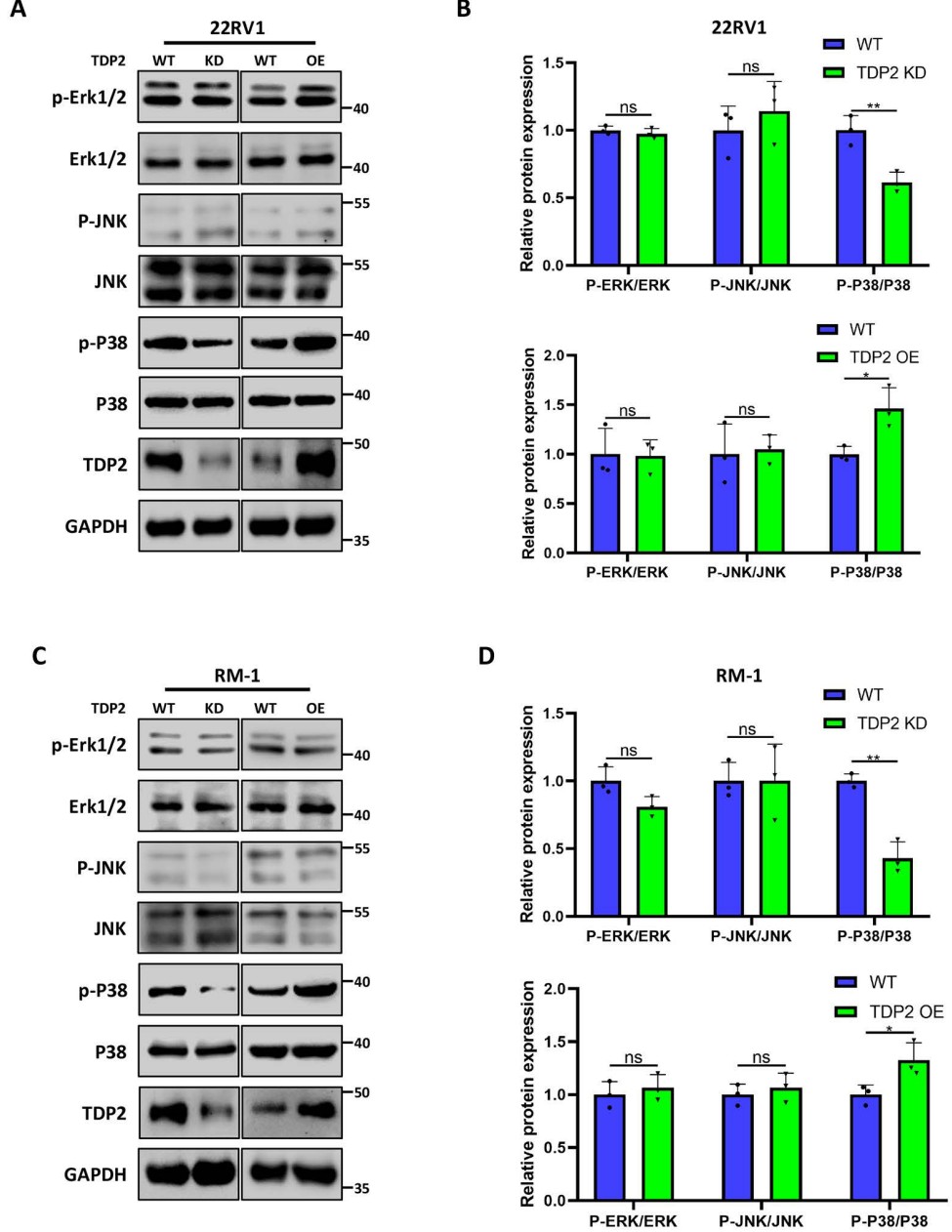

**Fig 4. TDP2 exerts biological functions through the MAPK pathway.** Western blot data demonstrating changes in protein concentration in 22RV1 cells transfected with shRNA-TDP2 or TDP2 **(A)**. Quantification of protein concentration in 22RV1 cells **(B)**. Western blot data demonstrating changes in protein concentration in RM-1 cells transfected with shRNA-TDP2 or TDP2 **(C)**. Quantification of protein concentration in RM-1 cells **(D)**. Error bars designate SD acquired from three independent experiments. P values were determined by the unpaired Student's t-test, ns Non-significant, *P<0.05, **P<0.01, ***P<0.001.

phosphorylation of JNK and ERK1/2(Figs 4A–4D and S5–S9). The phosphorylation of p38 was inhibited by inhibiting p38 (S3A, S3B, S10 and S11 Figs). Some literature reported p38α regulates the activation of cyclooxygenase 2 (COX2), which has a proinflammatory activity that has significant effects on cancer progression, such as nonmelanoma skin cancer, breast cancer [13,14]. It indicates the potential key role of TDP2 in PC immunosuppressive system.

## TDP2 promotes PC tumorigenesis in immunocompetent mice

We hypothesize that targeting TDP2 may have a role in reshaping the tumor immune microenvironment. In animal models, we further explored the impact of TDP2 on anti-tumor immunity in prostate cancer. TDP2 WT and TDP2 KD RM-1 cells were subcutaneously injected into immunocompetent C57BL/6J mice and immunodeficient nude mice, with tumor growth progress monitored regularly (Fig 5A and 5D). The results showed that TDP2 KD significantly inhibited tumor growth in mice, especially in immunocompetent C57BL/6J mice (Fig 5B, 5C, 5E and 5F). Immunohistochemical analysis revealed that TDP2 KD remarkably increased CD3 + T cell, CD8 + T cell, GZMB+ cell infiltration (Figs 5G, 5H, S3A and S3B). These findings suggest that the tumor-promoting effect of TDP2 is immune-dependent, implying that TDP2 may play a crucial role in shaping the immunosuppressive microenvironment in renal cancer.

## Cellular crosstalk in epithelial cells that highly express TDP2

Previous results have demonstrated the effects of TDP2-overexpressing epithelial cells 2 on the tumor microenvironment. These effects are likely to function through crosstalk between TDP2-overexpressing epithelial cells and other immune cells. Cell-cell communication with CellChat revealed that TDP2 high-expressing epithelial cells had significantly increased interactions with a variety of immune cells, including myeloid cells, monocytes, macrophages, and T cells, indicating that TDP2 high expression plays an important role in regulating immune responses in the tumor microenvironment (Fig 6A and 6B). This is highly consistent with the results obtained in our pathway enrichment analysis. This further confirms that epithelial cells with high expression of TDP2 2 may lead to the formation of an immunosuppressive microenvironment and epithelial-mesenchymal transition through interaction with myeloid cells and fibroblasts, thereby leading to poor tumor prognosis.

In addition, we found that TDP2 high-expressing epithelial cells 2 may exert their effects through COLLAGEN, GALECTIN, MK, and OSM signaling pathways in tumor progression. COLLAGEN, as a key component of the extracellular matrix, promotes the transition from epithelial to mesenchymal cells and enhances cell migration and invasiveness in the tumor microenvironment [15]. The interaction between TDP2 high-expressing epithelial cells 2 and fibroblasts may further increase the synthesis and degradation of collagen, exacerbating the invasiveness of the tumor (Fig 6C and 6E). GALECTINS, by modulating cell adhesion, migration, and immune evasion, suppress immune cell activity and promote tumor cell survival and metastasis [16]. Activation of the MK signaling pathway promotes tumor cell proliferation and migration, particularly in the interaction between TDP2 high-expressing epithelial cells and myeloid cells and fibroblasts, driving remodeling of the tumor microenvironment and invasiveness (Fig 6D, 6F, 6G and 6I). OSM, as a cytokine, regulates immune evasion and cell migration, enhancing tumor invasiveness and possibly promoting EMT [17], thereby facilitating metastasis (Fig 6H and 6J).

## The expression of TDP2 in prostate cancer cell inhibits immune cells activation in vitro

In this part, we performed a series of experiments to investigate whether TDP2 could inhibit antitumor immunity. DC and macrophages are both important innate immune cells [18]. Typically, macrophages can be classified as M1 or M2 macrophages [19]. The M1 subtype is the strong killer of cancer cells, while the M2 subtype is beneficial to sustaining immune suppressiveness [20,21].

 

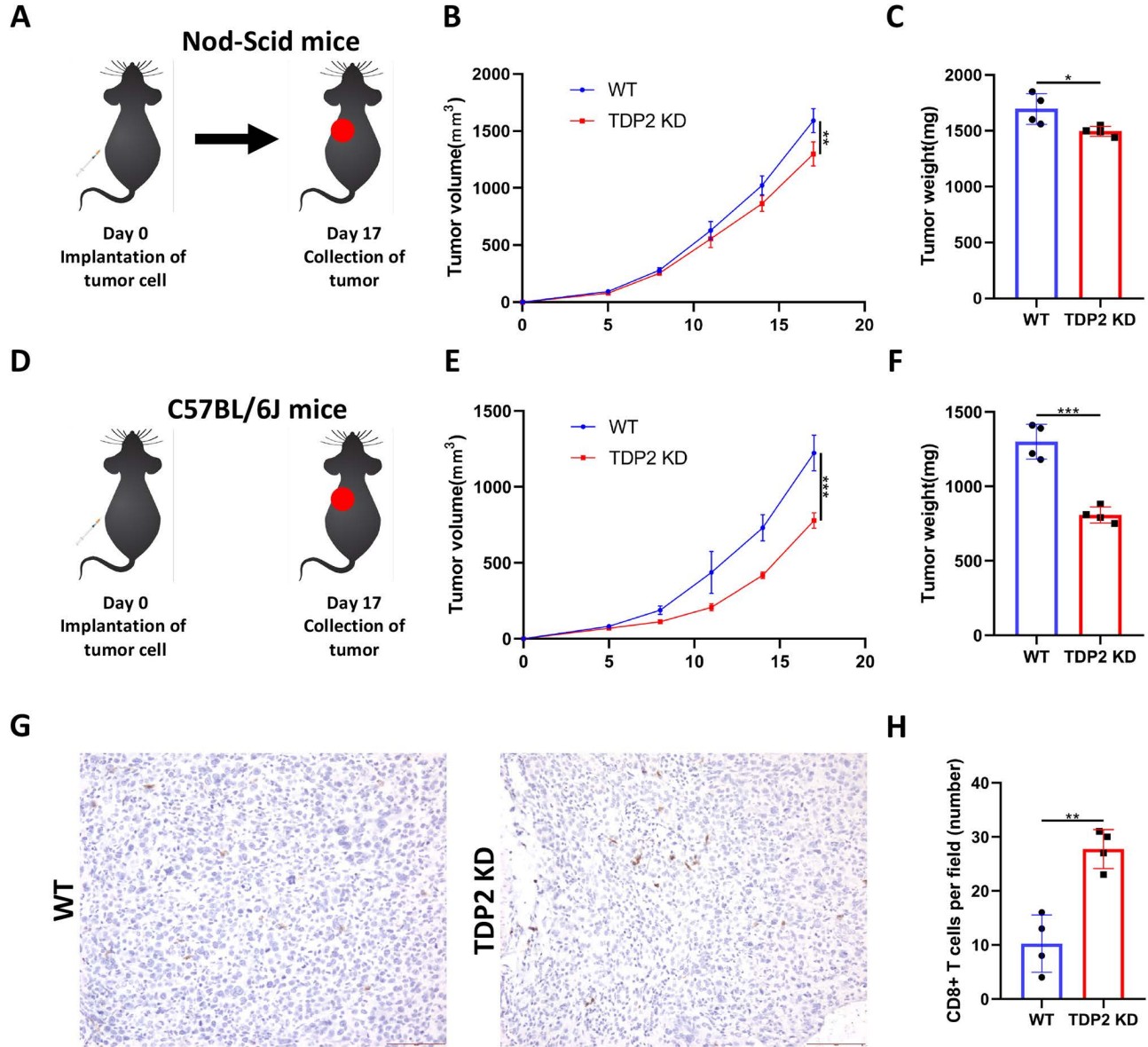

**Fig 5. TDP2 promotes tumor growth of RM-1 cell in vivo.** WT nude mice (n = 4) were Subcutaneously inoculated with WT RM-1 cells or TDP2 KD RM-1 cells **(A)**. Quantification of tumor growth **(B)** and tumor weight **(C)**. WT C57BL/6J mice (n = 4) were Subcutaneously inoculated with WT RM-1 cells or TDP2 KD RM-1 cells **(D)**. Quantification of tumor growth **(E)** and tumor weight **(F)**. Pathology studies show CD8 expression in the mice in different groups **(G)**. CD8 positivity analyses in the RM-1 model (n = 4) **(H)**. The bars represent 100 μm. Data were presented as means ± standard deviations. Error bars designate SD acquired from four independent experiments. P values were determined by the unpaired Student's t-test, ns Non-significant, **P < 0.01, ***P < 0.001.

Firstly, we examined whether the overexpression of TDP2 in RM-1 cell could inhibit the M1-polarization of macro-phages and the maturation of DCs. BMDMs and BMDCs were isolated from the bone marrow of mice in advance as previously described [22]. Lipopolysaccharide (LPS) can polarize macrophages and activate the immune response. Thus, treatments with LPS were conducted as a positive reference in the following experiments [23]. M1-like macrophage, M2-like macrophage, and mature DC were defined by CD86$^{high}$ MHC II$^{high}$ CD206$^{low}$, CD86$^{low}$ MHC II$^{low}$ CD206$^{high}$, and

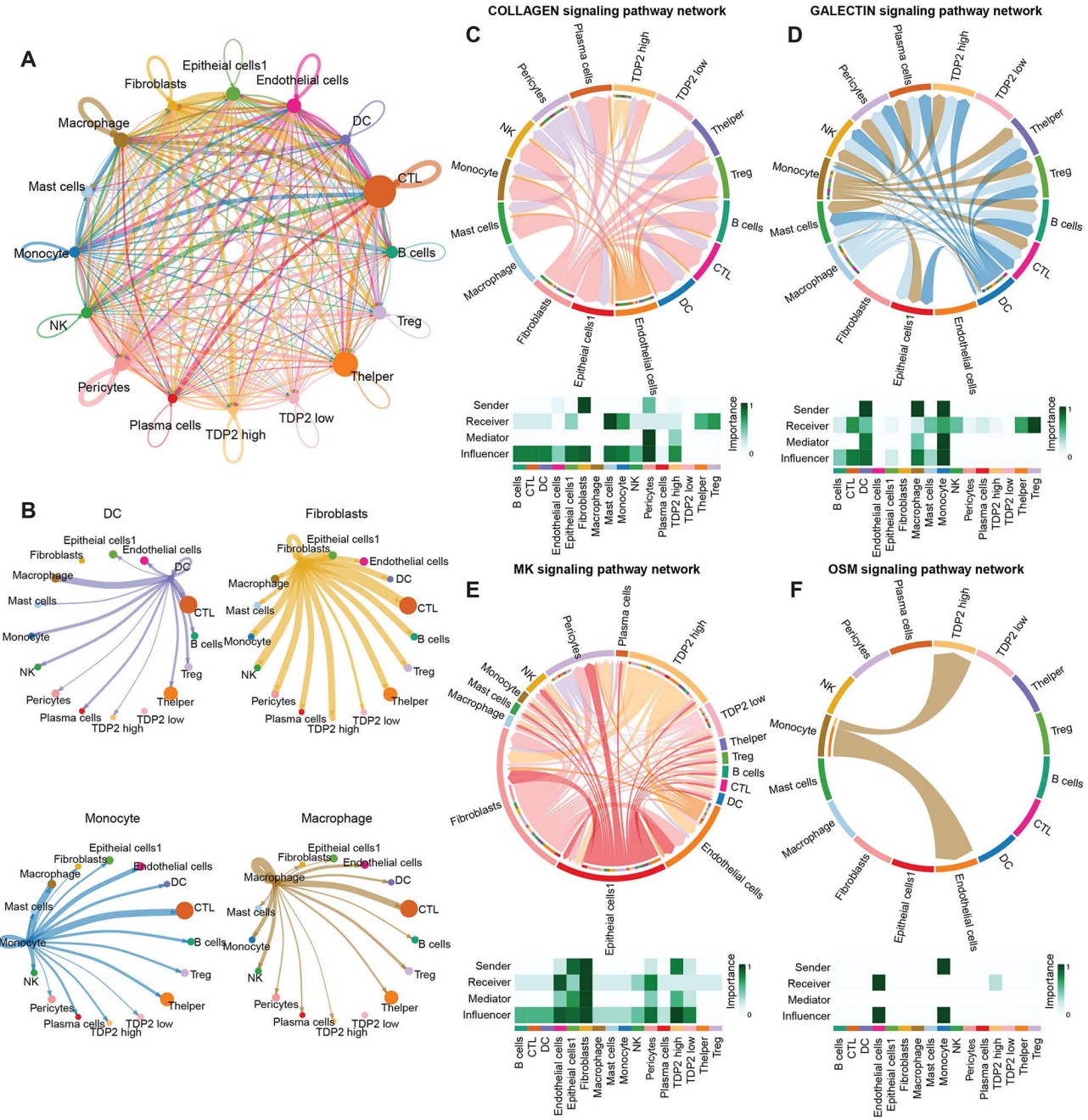

**Fig 6. Cell communications analysis results. A** Circular plot of the number of cellular interactions between each cell type. **B** Circular plot of the number of cellular interactions between myeloid cells, fibroblasts, and other cell types. **C**, **D**, **G**, **H** Chord diagram showing all the significant interactions (L–R pairs) between different cell types. Heatmaps of COLLAGEN **(E)**, GALECTIN **(F)**, MK **(I)** and OSM **(J)** signals contributing mostly to outgoing or incoming signaling of certain cell groups.

CD86^high, respectively. We observed that the overexpression of TDP2 in RM-1 cell could inhibit the M1-polarization of macrophages and the maturation of DCs (Fig 7A–7D). The impact of the expression of TDP2 in prostate cancer cell on T lymphocytes was studied subsequently. We concentrated on the CD8+ T cells which are recognized as the heart of adaptive

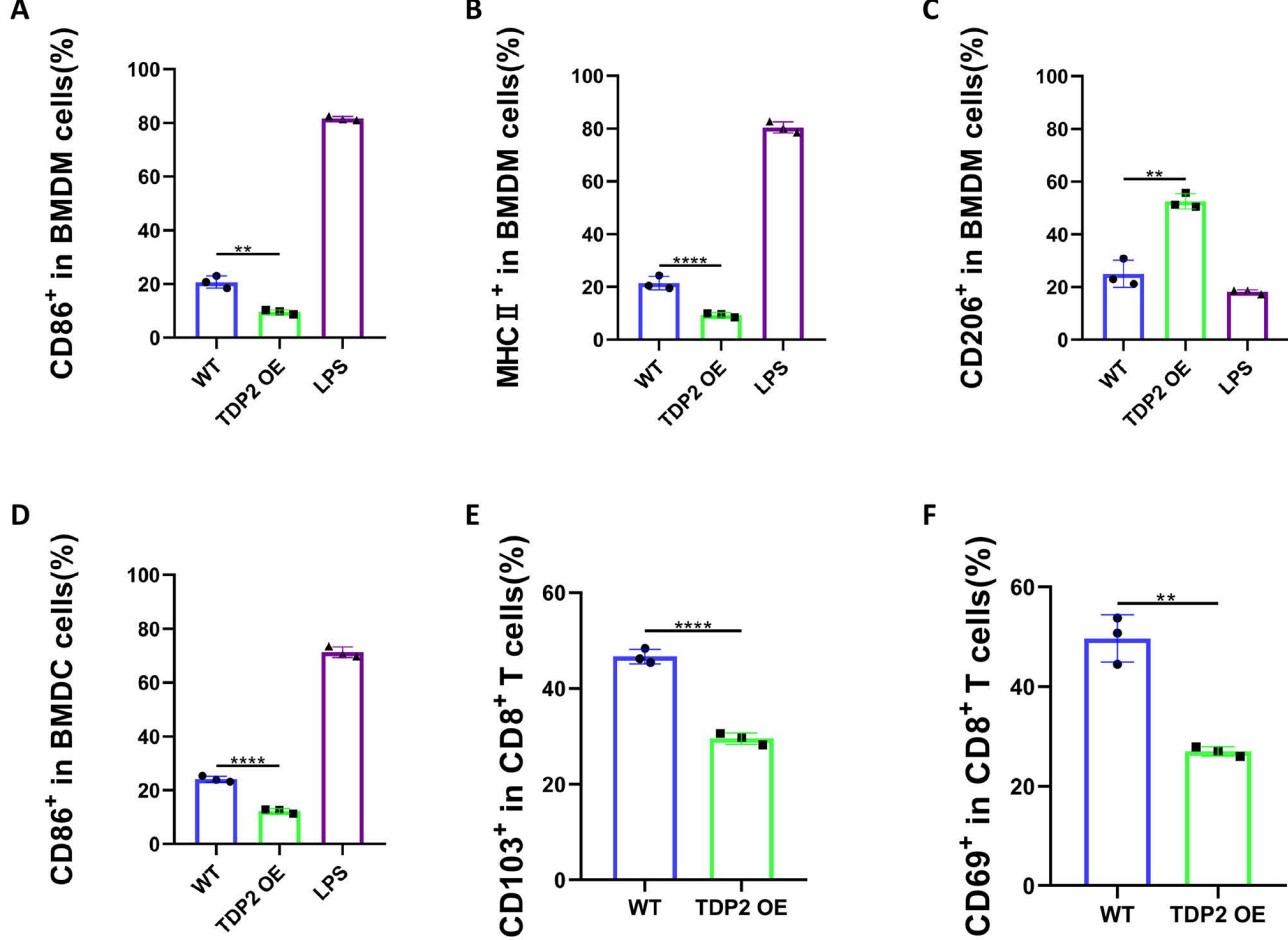

**Fig 7. In vitro immunosuppression of TDP2 overexpression.** The quantification of CD86+ macrophage **(A)**, MHC II+ macrophage **(B)**, and CD206+ macrophage **(C)** after co-culture in different groups. The quantification of mature DC cells after co-culture in various groups **(D)**. The quantification of CD103+ CD8+ T cells **(E)** and CD69+ CD8+ T cells **(F)** after co-culture in different groups. Data were presented as means ± standard deviations. Error bars designate SD acquired from three independent experiments. P values were determined by the unpaired Student's t-test, **$P < 0.01$, ****$P < 0.0001$.

immunity. We observed that the overexpression of TDP2 in RM-1 cell could inhibit the maturation of CD8+ T cells (Fig 7E and 7F). These results jointly revealed the expression of TDP2 in prostate cancer cell inhibits immune cells activation.

## Discussion

Our findings provide new insights into the role of TDP2 in prostate cancer, particularly in shaping the tumor microenvironment (TME) and its involvement in immune evasion and EMT. The analysis of single-cell RNA-seq data, in conjunction with pathway enrichment analysis, revealed that TDP2 high-expressing epithelial cells in the TME interact extensively with myeloid cells, macrophages, and fibroblasts, suggesting that TDP2 plays a crucial role in regulating immune responses and tumor progression.

One of the most striking findings of this study is the role of TDP2 high expression in modulating the immune response within the prostate cancer microenvironment. Cellular communication analysis revealed that TDP2 high-expressing epithelial cells exhibited significantly increased interactions with immune cells, including myeloid cells, macrophages, and T cells. This is consistent with our previous observation that TDP2 high cells promote immune evasion by inhibiting

macrophage polarization and dendritic cell (DC) maturation. Specifically, TDP2 overexpression in prostate cancer cells inhibited the M1 polarization of macrophages, which are typically associated with immune suppression and tumor progression [24]. Additionally, TDP2 high expression also suppressed CD8+T cell maturation, further enhancing the immune suppressive environment in the tumor microenvironment. These findings suggest that TDP2 could play a key role in regulating the immune microenvironment, facilitating immune evasion, and contributing to the tumor's ability to escape immune surveillance.

The relationship between TDP2 high expression and EMT was another focal point of our study. EMT is a well-known process involved in the acquisition of migratory and invasive properties by tumor cells, contributing to metastasis and therapy resistance. We observed that TDP2 high-expressing epithelial cells were enriched in signaling pathways associated with EMT—— COLLAGEN.

COLLAGEN, a major component of the extracellular matrix (ECM), was found to be involved in the transition from epithelial to mesenchymal cells, enhancing cell migration and invasiveness. Our study revealed that TDP2 high cells interact with fibroblasts to increase the synthesis and degradation of collagen, thereby promoting tumor cell invasion (Fig 6C and 6E). These findings align with previous studies showing that ECM remodeling, particularly collagen remodeling, is a critical driver of tumor metastasis in prostate cancer [25].

GALECTINS, including GALECTIN-1 and GALECTIN-3, are known to regulate immune cell adhesion, migration, and immune evasion [26]. In this study, TDP2 high expression was associated with upregulated GALECTIN signaling, further supporting its role in immune suppression and tumor progression. The MK signaling pathway was also enriched in TDP2 high-expressing cells, promoting tumor cell proliferation and migration, particularly through interactions with myeloid cells and fibroblasts. This pathway, involved in RNA metabolism and cell cycle progression [27], highlights TDP2's role in enhancing tumor aggressiveness. Finally, our analysis revealed that TDP2 high-expressing epithelial cells utilize OSM signaling to enhance immune evasion, cell migration, and EMT, facilitating tumor metastasis. As a cytokine, OSM plays a pivotal role in immune regulation and tumor cell migration, contributing to tumor progression [28].

In addition to its functional roles in immune modulation and EMT, our prognostic analysis indicated that TDP2 high expression is associated with poor clinical outcomes in prostate cancer patients. Lasso regression and Cox regression models highlighted TDP2 as a key prognostic biomarker, supporting its potential as a therapeutic target in prostate cancer. The Kaplan-Meier survival curve confirmed that TDP2 expression correlates with prognosis, where high expression of TDP2 is associated with shorter survival.

In summary, this study underscores the importance of TDP2 in shaping the tumor microenvironment in prostate cancer, particularly through its role in immune suppression and EMT. By interacting with fibroblasts, myeloid cells, and macrophages, TDP2 high expression enhances immune evasion and promotes tumor invasiveness through the COLLAGEN, GALECTIN, MK, and OSM signaling pathways. These findings provide new insights into TDP2's role in prostate cancer progression and highlight its potential as a prognostic biomarker and therapeutic target.

## Conclusions

In summary, this study identifies TDP2 as a critical regulator of the tumor microenvironment in prostate cancer. Through integrative analyses of single-cell transcriptomic data and functional validation, we demonstrate that TDP2-high epithelial cells interact extensively with immune and stromal cells, thereby promoting immune evasion and epithelial–mesenchymal transition. Mechanistically, TDP2 mediates these effects through activation of key signaling pathways, including COLLAGEN, GALECTIN, MK, and OSM. Functionally, TDP2 overexpression suppresses M1 macrophage polarization, dendritic cell maturation, and CD8+T cell activation, establishing an immunosuppressive milieu that favors tumor progression. Clinically, high TDP2 expression correlates with poor patient outcomes, highlighting its potential value as a prognostic biomarker and a therapeutic target in prostate cancer. These findings collectively provide new insights into the molecular mechanisms linking TDP2 to tumor progression and immune modulation.

## Supporting information

**S1 Fig. Biological functions of TDP2 expression in PC-3 and C4-2 cells.** Western blot data demonstrating changes in TDP2 protein concentration in PC-3 (A) and C4-2 (B) cells transfected with shRNA-TDP2 or TDP2. Cell growth curves (C, D) and colony formation assays (E, F) were performed to assess the proliferation of TDP2 KD or TDP2 OE transduced PC-3 and C4-2 cells. Quantification of the colony number in 22RV1 (G), RM-1 (H), PC-3 (I) and C4-2 (J) cells. Data were presented as means ± standard deviations. P values were determined by the one-way ANOVA, *P < 0.05, **P < 0.01, ***P < 0.001.
(PDF)

**S2 Fig. TDP2 expression increased migration abilities of 22RV1 and RM-1 cells.** Cell wound healing assays in 22RV1 (A) and RM-1 (B) cells with overexpression and knockout of TDP2. The width of cell wound healing was measured (n = 4/group). WT, wild type; TDP2 OE, TDP2 overexpression; TDP2 KD, TDP2 knockdown. Data were presented as means ± standard deviations. P values were determined by the one-way ANOVA, **P < 0.01.
(PDF)

**S3 Fig. P38 MAPK inhibitors suppressed TDP-induced p38 phosphorylation.** 22RV1(A) and RM-1 (B) WT/TDP2 OE cells were treated by 30 μM Adezmapimod. The expression of p38 and the corresponding phosphorylated forms was detected by Western blot.
(PDF)

**S4 Fig. Pathological test results analysis.** Pathology studies show CD3 (A) and GZMB (C) expression in mice across different groups. CD3 (B) and GZMB (D) positivity analysis in the RM-1 model (n = 4). The bars represent 100 μm. Data are presented as means ± standard deviations. Error bars indicate SD obtained from four independent experiments. P values were determined using the unpaired Student's t-test, ns = not significant, **P < 0.01, ***P < 0.001.
(PDF)

**S5 Fig. The original, uncropped gel of Fig 4A.**
(PDF)

**S6 Fig. The original, uncropped gel of Fig 4A.**
(PDF)

**S7 Fig. The original, uncropped gel of Fig 4C.**
(PDF)

**S8 Fig. The original, uncropped gel of Fig 4C.**
(PDF)

**S9 Fig. The original, uncropped gel of S1A and S1B Fig.**
(PDF)

**S10 Fig. The original, uncropped gel of S3A Fig.**
(PDF)

**S11 Fig. The original, uncropped gel of S3B Fig.**
(PDF)

**S1 Raw Image.**
(PDF)

## Author contributions

**Formal analysis:** Huan Cao, Anyin Pan, Yuetao Chen.

**Methodology:** Huan Cao, Anyin Pan.

**Project administration:** Huan Cao.

**Supervision:** Huan Cao.

**Validation:** Yuetao Chen, Fei Zheng.

**Visualization:** Yuetao Chen, Fei Zheng.

**Writing – original draft:** Huan Cao, Anyin Pan.

**Writing – review & editing:** Huan Cao.

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
