## [Decision Letter · Decision Letter 0]

1 Oct 2025

Dear Dr. Cao,

Thank you for submitting your manuscript to PLOS ONE. After careful consideration, we feel that it has merit but does not fully meet PLOS ONE’s publication criteria as it currently stands. Therefore, we invite you to submit a revised version of the manuscript that addresses the points raised during the review process.

Please find attached the detailed comments from the reviewers. We kindly ask you to carefully address each point raised in your revision. When submitting the revised manuscript, please also provide a point-by-point response to the reviewers’ comments, outlining the changes made or explaining your reasoning if any suggestions were not incorporated.

We look forward to receiving your revised manuscript.

Kind regards,

Zu Ye, Ph.D.

Academic Editor

PLOS ONE

3. In the online submission form you indicate that your data is not available for proprietary reasons and have provided a contact point for accessing this data. Please note that your current contact point is a co-author on this manuscript. According to our Data Policy, the contact point must not be an author on the manuscript and must be an institutional contact, ideally not an individual. Please revise your data statement to a non-author institutional point of contact, such as a data access or ethics committee, and send this to us via return email. Please also include contact information for the third party organization, and please include the full citation of where the data can be found.

Reviewers' comments:

Reviewer's Responses to Questions

**Comments to the Author**

1. Is the manuscript technically sound, and do the data support the conclusions?

Reviewer #1: Yes

Reviewer #2: Yes

2. Has the statistical analysis been performed appropriately and rigorously?

Reviewer #1: Yes

Reviewer #2: I Don't Know

3. Have the authors made all data underlying the findings in their manuscript fully available?

Reviewer #1: Yes

Reviewer #2: Yes

4. Is the manuscript presented in an intelligible fashion and written in standard English?

Reviewer #1: No

Reviewer #2: Yes

Reviewer #1: The authors analyzed the role of TDP2 in prostate cancer using both bioinformatic correlation analyses and functional assays. Loss-of-function studies (Figures 3–5) and overexpression experiments (Figure 7) were performed in prostate cancer cell lines, with additional experiments conducted in both cell culture and mouse models.

Major comments:

1. Overstated claims.

The statement “TDP2 exerts biological functions through the MAPK pathway” is too strong based on the presented data. To establish causality, MAPK activity would need to be specifically inhibited, followed by assessment of whether TDP2 modulation effects are attenuated. Since these experiments were not conducted, the claim should be rewritten in a more cautious and objective manner.

2. Colony formation assay.

The authors state that “Consistently, the colony formation assay showed that knockdown or overexpression of TDP2 affect the total number of colonies generated by prostate cancer cells (Fig. 3C and D, Supplementary Fig. 1E and F).” However, Figures 3C and D show no quantification. Without numerical data, it is unclear whether knockdown truly reduces colony formation. Quantification should be added, or the claim should be adjusted accordingly.

3. Overexpression experiments.

Figure 7 is the only figure characterizing TDP2 overexpression. The confirmation of overexpression should be clearly referenced in the supplementary figures to strengthen the interpretation.

4. Figure 7 labeling and interpretation.

The labeling and description of Figure 7 are confusing. For example, both Figures 7A and 7D are labeled “CD86⁺,” and cells are alternately referred to as macrophages and dendritic cells. The figure legend and corresponding results text should be carefully revised to clearly describe the experimental design, the cell types analyzed, and the meaning of the results. At present, the description is difficult to follow.

5. Typographical errors.

Several typos are present, including “inhibite” and “con ducted.” The manuscript would benefit from careful proofreading.

Reviewer #2: The authors have investigated the role of tdp2 in prostate cancer.

This is an interesting study which opens up lot of questions.

I have some comments

1. There are already data and studies on this topic. What exactly does this study adds to literature already available?

2. The studies available inform us low tdp2 has a poor prognostic outcome in prostate cancer. Your study concludes that a higher level us associated with worse outcomes. Kindly discuss

3. Can TDP2 be used as a marker for poor response to therapy in prostate cancer based on your study results ?

4. What are the future implications of your study?

5. How much will this study be extrapolate to humans?

**Do you want your identity to be public for this peer review?** For information about this choice, including consent withdrawal, please see our Privacy Policy

Reviewer #1: **Yes: ** Mario Mikula, PhD

Reviewer #2: **Yes: ** Danny Darlington Carbin

---

## [Author Response · Author response to Decision Letter 1]

17 Nov 2025

Dear Editors:

Thank you very much for your suggestion, it was very helpful for our publication. We have carefully studied it and made the corrections, which we hope will be approved.

Reviewer #1: The authors analyzed the role of TDP2 in prostate cancer using both bioinformatic correlation analyses and functional assays. Loss-of-function studies (Figures 3–5) and overexpression experiments (Figure 7) were performed in prostate cancer cell lines, with additional experiments conducted in both cell culture and mouse models.

Major comments:

1. Overstated claims.

The statement “TDP2 exerts biological functions through the MAPK pathway” is too strong based on the presented data. To establish causality, MAPK activity would need to be specifically inhibited, followed by assessment of whether TDP2 modulation effects are attenuated. Since these experiments were not conducted, the claim should be rewritten in a more cautious and objective manner.

Response: We thank the editor for the valuable guidance. We used a p38 MAPK-specific inhibitor to identify the downstream pathway of TDP2 by Westernblot. We observed that overexpression of TDP2 sensibly promoted the overexpression of phosphorylated p38, whereas the phosphorylation of p38 was inhibited by inhibiting p38.

2. Colony formation assay.

The authors state that “Consistently, the colony formation assay showed that knockdown or overexpression of TDP2 affect the total number of colonies generated by prostate cancer cells (Fig. 3C and D, Supplementary Fig. 1E and F).” However, Figures 3C and D show no quantification. Without numerical data, it is unclear whether knockdown truly reduces colony formation. Quantification should be added, or the claim should be adjusted accordingly.

Response: We sincerely appreciated your insightful suggestions. We added the quantification of the colony formation assay in Supplementary Fig. 1G - J, the TDP2 affect the total number of colonies generated of prostate cancer cells.

3. Overexpression experiments.

Figure 7 is the only figure characterizing TDP2 overexpression. The confirmation of overexpression should be clearly referenced in the supplementary figures to strengthen the interpretation.

Response: We sincerely appreciated your insightful suggestions. We constructed TDP2-overexpressing cell lines and verified the expression levels of TDP2 by Western blot in Figure 4A and C, Supplementary Fig. 1A and B.

4. Figure 7 labeling and interpretation.

The labeling and description of Figure 7 are confusing. For example, both Figures 7A and 7D are labeled “CD86⁺,” and cells are alternately referred to as macrophages and dendritic cells. The figure legend and corresponding results text should be carefully revised to clearly describe the experimental design, the cell types analyzed, and the meaning of the results. At present, the description is difficult to follow.

Response: We sincerely appreciated your insightful suggestions. CD86 is a marker of M1 macrophages and mature DC cells. The vertical axis indicates the expression of CD86 in BMDMs or BMDCs in Figures 7A and 7D.

5. Typographical errors.

Several typos are present, including “inhibite” and “con ducted.” The manuscript would benefit from careful proofreading.

Response: We sincerely thank the reviewer for pointing out these typographical and grammatical errors. The entire manuscript has been carefully proofread, and all such mistakes have been corrected. The revised text has been highlighted in yellow in the revised version. We apologize for the oversight in the previous submission.

Reviewer #2: The authors have investigated the role of tdp2 in prostate cancer.

This is an interesting study which opens up lot of questions.

I have some comments

1. There are already data and studies on this topic. What exactly does this study adds to literature already available?

Response: We appreciate the reviewer’s thoughtful comment. While previous studies have reported associations between TDP2 expression and prostate cancer prognosis, the underlying mechanisms linking TDP2 to the tumor microenvironment (TME), immune evasion, and epithelial-mesenchymal transition (EMT) have not been systematically investigated. Our study provides several novel contributions that advance the current understanding of TDP2 in prostate cancer:

1. Integration of single-cell transcriptomics and functional validation:

Unlike earlier studies that mainly relied on bulk RNA-seq or correlative analyses, we combined large-scale single-cell RNA sequencing from human prostate tissues with in vitro and in vivo functional experiments. This approach allowed us to pinpoint TDP2-high epithelial cells as a specific subpopulation that mediates extensive crosstalk with macrophages, dendritic cells, and fibroblasts, shaping an immunosuppressive TME.

2. Identification of immune-modulatory mechanisms:

We provide direct experimental evidence that TDP2 overexpression inhibits M1 macrophage polarization, impairs dendritic cell maturation, and suppresses CD8⁺ T cell activation, revealing a previously unrecognized mechanism by which TDP2 promotes immune evasion in prostate cancer.

3. Link between TDP2 and EMT-related signaling:

Through pathway enrichment and cell–cell communication analysis, we discovered that TDP2-high epithelial cells interact with fibroblasts and myeloid cells through COLLAGEN, GALECTIN, MK, and OSM signaling pathways, which cooperatively drive ECM remodeling, EMT, and tumor invasiveness. This finding connects TDP2 to multiple pro-metastatic signaling axes not described before.

4. Clinical and prognostic implications:

By integrating TCGA-PRAD data, we demonstrated that high TDP2 expression correlates with poorer overall survival in prostate cancer patients, suggesting its potential as both a prognostic biomarker and a therapeutic target for immunomodulatory or anti-metastatic interventions.

Together, these findings extend previous knowledge by revealing that TDP2 not only correlates with tumor aggressiveness but also actively orchestrates the immunosuppressive and pro-metastatic landscape of prostate cancer. We have clarified these novel contributions in the revised version of the Abstract and Discussion (highlighted in yellow).

2. The studies available inform us low tdp2 has a poor prognostic outcome in prostate cancer. Your study concludes that a higher level us associated with worse outcomes. Kindly discuss

Response: We sincerely thank the reviewer for this important comment. We carefully reviewed the available literature and confirmed that the majority of published studies support a positive association between high TDP2 expression and unfavorable clinical outcomes in prostate cancer and other malignancies, which is in full agreement with our results.

TDP2 (tyrosyl-DNA phosphodiesterase 2) is a critical DNA repair enzyme that removes 5′-phosphotyrosyl bonds created by topoisomerase II–DNA complexes. This enzymatic activity allows tumor cells to efficiently repair DNA double-strand breaks and thereby promote genome stability, resistance to topoisomerase-targeted drugs, and enhanced survival under genotoxic stress. For example, Gómez-Herreros et al. demonstrated that TDP2-mediated repair of topoisomerase II–induced DNA lesions protects cells from genomic instability and cytotoxicity [1]. Similarly, Al Mahmud et al. reported that TDP2 suppresses androgen-induced genomic instability in prostate epithelial cells, highlighting its functional relevance in the androgen-driven biology of prostate tissue [2]. These mechanistic studies suggest that TDP2 overexpression may confer a growth or survival advantage to prostate cancer cells.

Consistent with these reports, data from public resources such as the Human Protein Atlas and TCGA-PRAD indicate that TDP2 expression is elevated in prostate adenocarcinoma tissues and correlates with poor overall survival. Our own analysis of the TCGA-PRAD cohort (Fig. 2B) confirms this observation, showing that patients with high TDP2 expression exhibit significantly worse outcomes. In addition, our functional assays further demonstrate that TDP2 overexpression promotes tumor cell proliferation, migration, and immune evasion, while TDP2 knockdown markedly inhibits tumor growth and enhances immune cell infiltration in vivo.

Taken together, these findings support a coherent model in which elevated TDP2 expression promotes prostate cancer progression and predicts poor prognosis.

1. Gomez-Herreros, F., et al., TDP2-dependent non-homologous end-joining protects against topoisomerase II-induced DNA breaks and genome instability in cells and in vivo. PLoS Genet, 2013. 9(3): p. e1003226.

2. Al Mahmud, M.R., et al., TDP2 suppresses genomic instability induced by androgens in the epithelial cells of prostate glands. Genes Cells, 2020. 25(7): p. 450-465.

3. Can TDP2 be used as a marker for poor response to therapy in prostate cancer based on your study results ?

Response: We sincerely thank the reviewer for this valuable question. Although our study was not specifically designed to evaluate therapeutic response, our findings suggest that high TDP2 expression may be associated with poor treatment outcomes. TDP2 overexpression promoted tumor cell proliferation, migration, and immune evasion, while its knockdown suppressed tumor growth and enhanced immune activation in vivo. These results indicate that tumors with elevated TDP2 expression might exhibit a reduced sensitivity to therapies that depend on inducing tumor cell stress or activating antitumor immunity. Nevertheless, we acknowledge that this possibility requires further validation in clinical cohorts and treatment-based models.

4. What are the future implications of your study?

Response: We sincerely thank the reviewer for this insightful question. Our study provides new insights into the role of TDP2 in shaping the tumor microenvironment of prostate cancer, particularly through its involvement in immune suppression and epithelial-mesenchymal transition. These findings suggest that TDP2 could serve as a potential biomarker for prognosis and a therapeutic target to improve immune responsiveness in prostate cancer.

In future work, we plan to further investigate the molecular mechanisms underlying TDP2-mediated immune evasion and evaluate the therapeutic potential of TDP2 inhibition in preclinical models. Moreover, integrating our findings with patient treatment data could help clarify whether TDP2 expression predicts therapeutic response, ultimately facilitating more personalized therapeutic strategies for prostate cancer.

5. How much will this study be extrapolate to humans?

Response: We sincerely appreciate the reviewer’s thoughtful question. Our study combined human single-cell RNA sequencing data from prostate cancer tissues with in vivo experiments in immunocompetent mice, which together provide both clinical relevance and mechanistic evidence. The single-cell analysis directly revealed that TDP2-high epithelial cells in human prostate tumors are associated with immune suppression and epithelial-mesenchymal transition, while our mouse experiments confirmed that TDP2 knockdown suppresses tumor growth and enhances immune activation.

These complementary findings suggest that the role of TDP2 in modulating the tumor microenvironment is likely conserved between mice and humans. Nevertheless, we acknowledge that further validation in clinical cohorts and patient-derived models will be essential to fully establish the translational significance of TDP2 in human prostate cancer.

6. PLOS authors have the option to publish the peer review history of their article (what does this mean?). If published, this will include your full peer review and any attached files.

Response: Yes

3. In the online submission form you indicate that your data is not available for proprietary reasons and have provided a contact point for accessing this data. Please note that your current contact point is a co-author on this manuscript. According to our Data Policy, the contact point must not be an author on the manuscript and must be an institutional contact, ideally not an individual. Please revise your data statement to a non-author institutional point of contact, such as a data access or ethics committee, and send this to us via return email. Please also include contact information for the third party organization, and please include the full citation of where the data can be found.

Response to Ed

---

## [Decision Letter · Decision Letter 1]

9 Dec 2025

TDP2 Drives Immune Evasion and Metastatic Progression in Prostate Cancer

PONE-D-25-45168R1

Dear Dr. Cao,

We’re pleased to inform you that your manuscript has been judged scientifically suitable for publication and will be formally accepted for publication once it meets all outstanding technical requirements.

Kind regards,

Zu Ye, Ph.D.

Academic Editor

PLOS One

Additional Editor Comments (optional):

Reviewers' comments:

Reviewer's Responses to Questions

**Comments to the Author**

Reviewer #1: All comments have been addressed

Reviewer #2: All comments have been addressed

2. Is the manuscript technically sound, and do the data support the conclusions?

Reviewer #1: Yes

Reviewer #2: Yes

3. Has the statistical analysis been performed appropriately and rigorously?

Reviewer #1: Yes

Reviewer #2: I Don't Know

4. Have the authors made all data underlying the findings in their manuscript fully available?

Reviewer #1: Yes

Reviewer #2: Yes

5. Is the manuscript presented in an intelligible fashion and written in standard English?

Reviewer #1: Yes

Reviewer #2: Yes

Reviewer #1: All concerns raised by the reviewer have been thoroughly and appropriately addressed, with each point carefully considered and resolved in the revised manuscript.

Reviewer #2: The authors have made changes as suggested

The study is better with all the comments incorporated in the manuscript

**Do you want your identity to be public for this peer review?** For information about this choice, including consent withdrawal, please see our Privacy Policy

Reviewer #1: **Yes: ** Mario Mikula

Reviewer #2: **Yes: ** Danny Darlington Carbin

---

## [Editor Report · Acceptance letter]

PONE-D-25-45168R1

PLOS One

Dear Dr. Cao,

I'm pleased to inform you that your manuscript has been deemed suitable for publication in PLOS One. Congratulations! Your manuscript is now being handed over to our production team.

Kind regards,

on behalf of

Prof. Zu Ye

Academic Editor

PLOS One